# Dermo-Seg: ResNet-UNet Architecture and Hybrid Loss Function for Detection of Differential Patterns to Diagnose Pigmented Skin Lesions

**DOI:** 10.3390/diagnostics13182924

**Published:** 2023-09-12

**Authors:** Sannia Arshad, Tehmina Amjad, Ayyaz Hussain, Imran Qureshi, Qaisar Abbas

**Affiliations:** 1Department of Computer Science, Faculty of Basic and Applied Science, International Islamic University, Islamabad 44000, Pakistan; sannia.phdcs137@iiu.edu.pk (S.A.); tehminaamjad@iiu.edu.pk (T.A.); 2Department of Computer Science, Quaid e Azam University, Islamabad 44000, Pakistan; ayyaz.hussain@qau.edu.pk; 3College of Computer and Information Sciences, Imam Mohammad Ibn Saud Islamic University (IMSIU), Riyadh 11432, Saudi Arabia; iqureshi@imamu.edu.sa

**Keywords:** skin lesion disease, convolutional neural networks, UNet model, loss function, transfer learning, semantic segmentation, detection of patterns

## Abstract

Convolutional neural network (CNN) models have been extensively applied to skin lesions segmentation due to their information discrimination capabilities. However, CNNs’ struggle to capture the connection between long-range contexts when extracting deep semantic features from lesion images, resulting in a semantic gap that causes segmentation distortion in skin lesions. Therefore, detecting the presence of differential structures such as pigment networks, globules, streaks, negative networks, and milia-like cysts becomes difficult. To resolve these issues, we have proposed an approach based on semantic-based segmentation (Dermo-Seg) to detect differential structures of lesions using a UNet model with a transfer-learning-based ResNet-50 architecture and a hybrid loss function. The Dermo-Seg model uses ResNet-50 backbone architecture as an encoder in the UNet model. We have applied a combination of focal Tversky loss and IOU loss functions to handle the dataset’s highly imbalanced class ratio. The obtained results prove that the intended model performs well compared to the existing models. The dataset was acquired from various sources, such as ISIC18, ISBI17, and HAM10000, to evaluate the Dermo-Seg model. We have dealt with the data imbalance present within each class at the pixel level using our hybrid loss function. The proposed model achieves a mean IOU score of 0.53 for streaks, 0.67 for pigment networks, 0.66 for globules, 0.58 for negative networks, and 0.53 for milia-like-cysts. Overall, the Dermo-Seg model is efficient in detecting different skin lesion structures and achieved 96.4% on the IOU index. Our Dermo-Seg system improves the IOU index compared to the most recent network.

## 1. Introduction

Skin cancer is regarded as one of the most perilous kinds of cancer amongst the various categories. Its occurrence is particularly prevalent in the United States, with nearly one million positive diagnoses annually. There exist numerous varieties of skin cancers, and melanoma stands out as one of the most menacing types. According to the American Cancer Society, melanoma results in over 9000 fatalities and accounts for 76,380 new cases each year [1]. Each type of skin lesion possesses distinct structures or characteristics. The presence of these structures plays a key role in determining the specific form of skin cancer. These attributes encompass features such as negative networks, pigment networks, streaks, globules, milia-like cysts, cobblestones, hairpin-like structures, and more [2]. While evaluating skin lesions in clinical settings, dermatologists not only consider the typical and local appearance of a skin lesion but also examine these features in order to make informed decisions regarding the lesion’s type and level of malignancy. Figure 1 and Figure 2 provide examples of such attributes.

Dermatologists rely on the presence of these attributes to accurately identify and diagnose the specific type of skin lesion, along with its relevant contextual information. However, unfortunately, there are multiple challenging factors [2] associated with these attributes. Among these factors, the most significant one is the high degree of resemblance observed among various types of skin attributes [3]. This resemblance can manifest via several aspects, such as appearance and color of the lesion, as well as its location and the structures and texture therein. Consequently, this challenge may pose obstacles and create confusion for dermatologists in accurately determining the type of a skin lesion. Moreover, it can introduce uncertainties when trying to diagnose the presence of an attribute that plays a crucial role in causing a particular type of skin lesion.

Another challenge arises from the fact that the appearance of skin lesions can change depending on the type of lesion and the severity of the malignancy [3]. While some lesions may lack specific dermoscopic patterns, others may exhibit textural patterns associated with various attributes [3]. The automated assessment of dermoscopic images faces additional obstacles due to artifacts present in the setup of image capture, samples preparations, and the inherent appearance of the lesion itself. When acquiring images of the samples of a skin lesion via dermatoscope, various artifacts can impact the quality of the output. These artifacts may include color, gel bubbles, hair occlusion, low image contrast, ruler markers, uneven illumination, clothing obstacles, and lens artifacts, and marker annotations [4]. Furthermore, skin lesions can display significant inconsistencies in appearance depending on their type, with severe cases exhibiting fuzzy borders, indistinct texture, low contrast beside the surrounding skin, shapeless geometry, and multicolored characteristics.

Detecting attributes within skin lesions is also challenging due to highly imbalanced class distribution, a lack of sufficient samples, and significant intra-class and low inter-class variation. Therefore, there is a pressing need to study the various attributes present in skin lesions. In-depth exploration and identification of these features will contribute to the interpretability of models, enabling dermatologists to analyze the attributes that play a crucial role in identifying types of skin lesion.

The visual detection of attributes in skin lesions by humans is often difficult and time-consuming. Consequently, a computer-aided diagnostic (CAD) system is necessary to help dermatologists in analyzing these attributes. Numerous diagnostic techniques, including CAD systems, have been proposed in the literature for skin lesion segmentation [5,6,7,8,9,10,11,12,13,14,15,16,17,18,19,20,21,22,23,24,25,26,27,28,29,30,31,32,33,34,35,36,37,38,39,40,41,42]. However, many of these techniques primarily focus on boundary segmentation [5,6,7,8,9,10,11,12,13,14,15,16,17,18,19,20,21,22,23,24,25,26,27,28] while giving less attention to the segmentation of actual attributes within the lesion [29,30,31,32,33,34,35,36,37,38,39,40,41,42]. The purpose of employing CAD systems is to automatically identify irregularities or skin disorders present within the lesion.

According to recent research, incorporating lesion attribute segmentation into computer-aided diagnostic (CAD) systems can lead to a noteworthy enhancement in system performance. Deep learning methods, such as F-Net [43] and U-Net [44], are commonly employed for object detection and segmentation, including skin lesions segmentation. Most of the methods utilize ensemble strategies with different pre-trained backbones. In the winning method [31] of the ISIC 2018 contest, four pre-trained backbone networks (ResNet152 [45], DenseNet169 [46], Xception [47], and ResNetV2 [48]) were utilized in the encoder part of U-Net [43] segmentation models, followed by transfer learning on the dataset of the ISIC 2018.

However, there are only a few existing methods discussed in the literature on attribute segmentation in skin lesions [29,30,31,32,33,34,35,36,37,38,39,40,41,42]. Current skin lesion segmentation methods typically follow a traditional approach, involving the detection of the lesion area in the image, the extraction of discriminative features from the lesion, and the subsequent classification of the skin lesion type. CNN-based models can extract pertinent structures from dermoscopic images and classify them without utilizing attribute segmentation information [49]. Incorporating more information, such as attribute segmentation maps, can enhance the lesion classification task [50,51]. Yu et al. [50] demonstrated the effectiveness of incorporating lesion masks in elevating lesion classification, achieving the top rank in the ISIC 2016 contest. Similarly, Gonzalez Diaz [52] presented a skin lesion classification method that combined both lesion segmentation and attribute probability maps, achieving desirable results using a single model. These research efforts emphasize the importance of lesion attribute segmentation for diagnostic purposes and contribute to enhancing the interpretability of the model.

Additionally, it is crucial to include localization information in the output. Determining which structure and spatial location in the image contribute most to the identification and classification of the skin lesion type and its structure is essential. In previous competitions, it has been observed that there is significantly rare participation in the second task of the ISIC 2018 [52], which involves lesion pattern prediction, compared to other tasks [52]. This indicates that the second task is particularly challenging, prompting a focus on this problem instead of the others. Attribute segmentation can provide visual interpretation of structures that dermatologists can visually examine and investigate, helping to achieve more accurate and precise identification of whether a lesion is benign or malignant.

The ISIC 2018 dataset [52] provides labels for five interesting lesion attributes: globules, pigment networks, negative networks, streaks, and milia-like cysts, which are used for training [53]. Some of these attributes may appear on only one super-pixel, while others may cover most of the lesion area. Furthermore, only the attributes within the most prominent lesion are labeled. Our focus in this study is Task 2 of the ISIC 2018 dataset [52], which involves predicting the locations of dermoscopic attributes/structures in an image. By identifying these significant skin lesion structures, we can detect abnormal regions and provide explanations for practitioners to confirm and provide additional diagnoses.

However, the ISIC-2018 dataset [52] suffers from a scarcity of training data, and there is an imbalance in the number of samples for each structure. Legal restrictions and limited access to dermatology-related datasets pose challenges in acquiring the required data for developing statistical models with which to address dermatology-related issues. Moreover, openly accessible data, when accessible, are often very limited, encompassing only a few dozen images per class. The manual marking of these images is a slow process that requires the involvement of specialists, making it costly. Hence, there are challenges in obtaining further data for assessment and in efficiently utilizing existing data. Currently, there are very few deep learning-based approaches [30,31,32,33,34,35,36,37,38,39,40] employed for the segmentation of given lesion structures, such as in Task 2 of the ISIC 2018 Challenge [52].

To tackle these issues, we propose a solution for the attribute segmentation of skin lesions in this paper. We utilize the UNet segmentation model [44] with a ResNet-50 [45] backbone to perform attribute- and structural-level segmentation and localization of skin lesion attributes within an image. The main contributions of our proposed Dermo-Seg model are as follows:Dermo-Seg model is proposed consisting ResNet-50 as backbone architecture in the encoder part of UNet model to down-sample the input features of skin lesions. The purpose of down-sampling is to extract the high level feature maps of lesion’s area.These high-level feature maps contain very useful information about the attributes of skin lesions, which is then up-sampled in the decoder part of UNet and further concatenated with the high-level feature maps from the encoder part.A hybrid loss function, a blend of Focal Tversky loss [54] and mean IOU loss [55], is used to address the problem of class imbalance in the ISIC 2018 Task 2 dataset.Individual models are proposed for all five separate attributes of skin lesions.The hybrid loss function resolves the issue of class imbalance at the pixel level for each individual attribute.

The proposed approach enhances our ability to detect the presence of structures. By identifying various structures such as globules, streaks, milia, pigment networks, etc., within a lesion, we can gain knowledge surrounding the lesion type more effectively.

It is important to note that this work has certain limitations. We faced the challenge of not having comprehensive information about both the skin lesion type and its structural characteristics within one dataset. The available benchmarked ISIC dataset does not provide this combined information in a single dataset. Unfortunately, we are only able to detect attributes present in the skin lesion and not the specific type of skin cancer.

Dermoscopy allows for the visualization of structures and patterns [2] that are not detectable by naked eye, enabling dermatologists to make more accurate assessments. Analyzing these structures [2] plays a crucial role in the diagnosis and classification of skin cancer lesions.

In what follows, we discuss the key dermoscopy differential structures commonly observed in skin cancer lesions and their significance in clinical practice:(1)Pigment Network: The pigment network is one of the fundamental dermoscopic structures observed in skin cancer lesions [2]. It refers to the regular or irregular distribution of pigmented lines and grids within the lesion. The evaluation of the pigment network can provide insights into the organization and pattern of pigmented cells, which aids in distinguishing benign lesions from malignant ones. Irregular or disturbed pigment network patterns are mostly an indication of the presence of malignancy. It is the most important structure in the diagnosis of melanoma. The assessment of the pigment network involves assessing its distribution, thickness, regularity, and overall arrangement. Differences from a regular, symmetrical pattern may give an indication of malignancy.(2)Globules: Globules are-round shaped structures seen within the skin lesion [2]. They may vary in size, color, and distribution. The deep assessment of globules assists in discriminating between benign and malignant lesions. This assessment involves assessing their size, shape, color uniformity, and arrangement. In most of the benign lesions, globules are usually regular, symmetric, and evenly distributed. Malignant lesions mostly exhibit irregular, asymmetrical, or clustered globules, indicating of malignancy. Globules attributes can be pigmented or non-pigmented or both and may provide valuable information for differentiating melanoma from benign lesions.(3)Streaks: Streaks are commonly linear structures mostly found within the lesion [35]. These attributes can give valuable diagnostic information. In benign lesions, streaks are uniform most of the time, thin, and evenly distributed. On the other hand, malignant lesions may show thick, irregular, and asymmetric streaks, which can indicate malignancy. Moreover, the presence of a radial streaking pattern, moving outward from the center of the lesion, may be a strong indicator of invasive melanoma.(4)Negative Network: The negative network, also called the hypopigmented network [35], appears as lighter or sometimes colorless lines or areas within the lesion. It shows the absence of pigmentation in various areas. The presence of a negative network may indicate various types of melanoma or other non-melanocytic skin lesions.(5)Milia-like Cysts: milia-like cysts are small and white cyst structures [35] found within the skin lesion. These structures resemble milia, which are tiny epidermal cysts commonly seen in healthy skin. In the context of skin cancer, the presence of milia-like cysts can be indicative of specific subtypes or serve as a diagnostic clue for differentiating benign and malignant lesions.

Skin lesion classification relies heavily on the impactful and informative features extracted from accurately segmented skin lesions. Several techniques, based on traditional machine learning models, have been suggested in the literature [5,6,17,28,37,38,39,40] for the segmentation of skin lesions. However, these techniques solely emphasise the boundary segmentation rather the than structural segmentation of skin lesions. One such segmentation method, proposed by S. Garg and B. Jindal [5], consists of pre-processing and segmentation steps. The method employs thresholding and morphological operations to minimize artifacts. It utilizes a hybrid of K-means and firefly algorithms (FFA) to achieve skin lesion segmentation. K-means is employed to identify the precise lesion region, followed by FFA for optimizing clustering and enhancing accuracy. The method’s performance is assessed using ISIC 2017 [53] and PH2 datasets, achieving an accuracy rate of 99.1% and 98.9%, respectively.

N. Durgarao et al. [6] proposed a three-step technique to segment skin lesions; these steps include segmentation, feature extraction, and classification. Fuzzy C-means (FCM) clustering [56] is used for segmentation, while features are extracted using a hybrid of local vector pattern (LVP) [57] and local binary pattern (LBP) [58], which serves as an input for a fuzzy classifier. The significant contribution of this technique lies in its optimizing of the member function in the fuzzy classifier through the introduction of the DOROA algorithm. H. M Unver et al. recommended a mixture of the CNN model “YOLO” [59] and the grabcut algorithm [17] for skin lesion segmentation. Their algorithm operates in four steps: hair removal on the lesion, lesion detection, lesion segmentation, and post-processing. The working of the model was assessed on the PH2 dataset and ISIC 2017 [53], where it demonstrated good results on the ISIC dataset.

N. Fulgencio et al. proposed another model [28] for the segmentation of skin lesions which utilizes an innovative adaptation of superpixel techniques. This model yielded satisfactory results on the dataset of ISIC 2017. Additionally, they introduced a modified image registration approach to determine the features explaining the skin lesion and to capture various phases of the same lesion using two images. Talha et al. introduced a threefold approach [37] for the detection and classification of skin lesions, employing segmentation and feature selection techniques. They utilized three color spaces to separate the foreground image from the background and introduced a weighting criterion to choose the appropriate solution based on extensive structure analysis and central distance. Furthermore, they introduced an improved method of feature mining and dimensionality reduction, which outperformed existing techniques. R. Kaur et al. presented fifteen thresholding methods [38] for the segmentation of basal cell carcinoma (BCC), along with two error metrics: relative XOR error and the ratio for lesion capture. The geodesic active contour technique [39] was used for the segmentation of skin lesions, providing accurate contours by automatic initialization and overcoming issues arising due to noise, including hair. These techniques performed well in the presence of various artifacts, variations in structure, weak boundary strength, color, and structure. The borders were recreated using border smoothing, the calculation of spectral difference modification in Otsu thresholding, and inlet removal.

Chun-yan Yu et al. [40] offered an active contour model using a level set approach, which includes region information to obtain image contours. They applied distance regularization to penalize deviations of the level set from the signed distance function. Synthetic and real image data were used for experimentation. Their results showed the efficacy of their proposed model.

Various skin lesion segmentation approaches using deep learning have been proposed in the literature. One method [41] suggests an initial contour for recognizing the boundary of the skin lesion using the genetic algorithm to optimize the edge Chan-Vese method. Another method [42] utilizes a fully convolutional encoder–decoder architecture that is optimized with the exponential neighborhood grey wolf optimization algorithm for dermoscopic sample segmentation. A novel CNN-based architecture [7] called end-to-end atrous spatial pyramid pooling is designed for lesion segmentation. The system [8] combines Retina-DeepLab, graph-based techniques, and Mask R-CNN [60] for the segmenting of skin lesions. A dense encoder–decoder framework [9] combines ResNet and DenseNet, incorporates ASPP for multiscale contextual information, and uses skip connections for information recovery.

An automatic technique for skin lesion segmentation [10] utilizes an adaptive dual attention component using three properties: integration of ADAM with two global context modeling schemes, support for multi-scale fusion, and the spatial information weighted technique for redundancy reduction. An improved fully convolutional network approach [11] achieves skin lesion segmentation without pre-processing or post-processing by identifying the center position of the skin lesion and clearing edge details. A segmentation technique known as FC-DPN [12], combining the fully convolutional and dual path networks, automatically segments skin lesions. An attentive border-aware system [13] employs adversarial schooling, ResNet34, Scale Att-ASPP, and PPM for segmenting multi-scale lesions.

A Mask R-CNN-based technique [14] proposes candidate object bounding boxes and binary mask prediction for skin lesion segmentation. A combination of YOLOv3 and the GrabCut algorithm [15] is utilized for effective lesion segmentation. A lesion segmentation methodology inspired by the pyramid scene-parsing network [16] uses an encoder–decoder system with pyramid pooling blocks and skip connections. DeepLabv3+ and Mask R-CNN [18] are used for accurate lesion localization. Deep learning methods [19] combine pre-trained VGG16 [61], DeeplabV3 [62], SegNet decoder [63], and TernausNet [64] for skin lesion segmentation. U-net frameworks [20] and a modified U-Net framework obtain an effective lesion segmentation. A dense deconvolutional framework [21] addresses the challenges of changing size and appearance in skin lesion segmentation.

Along with the above given techniques, there is very little literature concerning the attribute-level segmentation of skin lesions [29,30,31,32,33,34,35,36,37,38,39,40,41,42]. Nguyen D et al. [29] proposed the TATL method, an innovative transfer learning approach for detecting skin attributes. Through massive experiments on the ISIC 2018 and ISIC 2017 datasets, the efficacy of TATL has been validated. Their proposed model outperformed existing methods while utilizing only 1/30th of the parameters compared to the winner of the ISIC2018 competition. Notably, TATL demonstrates significant improvements in the diagnosis of skin lesion methods that are pre-trained on ImageNet, particularly for structures with limited training samples. TATL [29], which stands for “task-agnostic transfer learning”, is inspired by the behaviors of dermatologists in the context of skincare. It utilizes an attribute-agnostic segmenter that learns to detect regions of skin lesions. After that, it transfers the obtained knowledge to a set of attribute-specific classifiers for detecting specific attributes. The attribute-agnostic segmenter in TATL focuses solely on identifying skin attribute regions, which allows it to leverage abundant data from all attributes, facilitate knowledge transfer among features, and compensate for the limited training data available for rare attributes.

Another method for the structure segmentation of skin lesion is proposed by Kadir M. et al. [30]. The authors introduced a method for CNN fine-tuning that allows users to provide feedback related to two aspects simultaneously: the classification and visual description accompanying the classification. In their study, they focused on the classification of skin lesion and examined how CNNs respond to these two user feedbacks. They proposed a new CNN architecture that incorporates the Grad-CAM [65] technique to explain the model’s decision during the training process. Using user feedback, the authors discovered that fine-tuning their model based on classification and description leads to improved visual explanations while maintaining classification accuracy. This finding has the potential to enhance user trust in CNN-based skin lesion classifiers as users can better understand and interpret the model’s decisions through the accompanying visual explanations.

One more novel approach is given by Nguyen D et al. [31] for the automated prediction of the places of dermoscopic structures in Task 2 of the ISIC 2018 Challenge. This method uses the Attention U-Net model and multi-scale images as input. To improve the performance, the authors employ a transfer learning strategy by adjusting the weights of a pre-trained network used for segmentation to train the deep neural network for feature extraction. Their proposed algorithm achieves comparable performance to that of LeHealth [66] and NMN [31].

Another segmentation model was intended by Jahanifar M et al. [32]. It incorporates transfer learning to segment lesions and their structures using convolutional neural networks (CNNs). The model adopts an encoder–decoder architecture, leveraging various pre-trained models in the encoding path. It produces prediction maps by merging multi-scale information in the decoding path through a pyramid pooling approach. To overcome the challenge of limited training data and enhance the generalization of their model, a comprehensive set of novel domain-specific augmentation techniques was employed. These techniques simulate real variations found in dermoscopy images. Their method achieved the top ranking on the leaderboard for the ISIC2018 attribute detection task.

A deep learning approach called the superpixel attention network (SANet) was proposed [33] in the literature. In this framework, input images are segmented into small regions, which are then shuffled using the random shuffle mechanism (RSM). Subsequently, the SANet is applied to capture distinct features and recreate the input images. The SANet model comprises two sub-modules: superpixel average pooling and superpixel attention module. A superpixel average pooling technique is introduced to redevelop the problem of superpixel classification into a superpixel segmentation problem. The superpixel attention module (SAM) [67] is employed to emphasize discriminative superpixel regions and feature channels. To handle the issue of serious data imbalance in the ISIC 2018 Task 2 [52], the researchers proposed a loss function called the global balancing loss. The proposed method demonstrated better performance on the ISIC 2018 Task 2 challenge [52].

There is another method in which researchers [35] devised a comprehensive solution for detecting dermoscopic attributes by utilizing the ISIC competition dataset. They explored the potential for enhancing performance by incorporating labels of various modalities in the form of segmentation masks. To achieve this, they proposed leveraging the task of segmentation as an auxiliary task and enabling knowledge sharing between the two tasks. They accomplished this by training a Y-net neural network architecture, which facilitated weight sharing between the segmentation and classification. Furthermore, they put forward multiple approaches for combining the training of Y-Net using segmentation and classification labels.

Kawahara J et al. [36] reframe the objective of categorizing clinical dermoscopic attributes within super-pixels by handling it as a segmentation challenge. They put forth a fully convolutional neural network that is specifically designed to identify these dermoscopic attributes in dermoscopy images. The architecture of their neural network incorporates interposed feature maps derived from multiple middle layers of the network. To tackle the issue of imbalanced labels, they employ a negative multilabel Dice-F1 score as a loss function, which calculates the score across the minibatch for each label and minimizes it.

A framework that incorporates transfer learning into the segmentation of skin cancer lesions and their structures using CNNs was proposed by Koohbanani N et al. [37]. This framework draws inspiration from the widely recognized UNet architecture. In the encoding path, they leverage various pre-trained networks; while in the decoding path, the pyramid pooling approach is utilized to combine multi-scale information and generate the prediction map. To overcome the limited accessibility of training data and enhance the generalization of model, a diverse set of novel augmentation techniques was employed during network training. The authors designed a loss function to handle the challenges encountered through the training phase.

Nunnari F et al. [38] investigated the correlation between visual features and areas recognized by CNNs learned for classification. Through experiments with two different neural network architectures with varying deepness and resolution in the last convolutional layer, they quantified the effectiveness of threshold Grad-CAM saliency maps in identifying visual features associated with skin cancer.

Gonzalez Ivan [39] integrated the expertise of dermatologists into the widely used framework of CNNs. The approach involved designing multiple networks that incorporate lesion area recognition in the segmentation of lesions into attribute patterns. Additionally, the author has developed innovative blocks for CNNs to seamlessly integrate this information into the diagnosis processing pipeline.

A multi-task U-Net model for the automatic detection of melanoma lesion attributes was proposed in [41]. This model incorporates two tasks: classification, to determine the presence of lesion attributes; and segmentation, to identify and delineate the attributes within the images. The multi-task U-Net model achieved a Jaccard index of 0.433 on the test data of the ISIC 2018 Challenges Task 2. This method secured the 5th position in the final leaderboard of the ISIC 2018.

Labraca J et al. [42] presented an approach to enhancing the diagnostic process for doctors. It leverages dermoscopic-structure-based soft segmentation to develop a set of classifiers specific to different dermoscopic structures. Each classifier focuses on distinguishing between benign lesions and melanomas based on a particular dermoscopic structure. The outputs of these individual classifiers are then combined using the Bayesian method. This not only provides the final diagnosis but also offers valuable additional information to the doctor. This includes the opinions of the individual structure-specific experts and the uncertainty associated with the diagnosis, enriching the diagnostic process. A comparison of the state-of-the-art methods for segmenting skin lesion attributes is given in Table 1.

For all we know, the existing literature lacks satisfactory performance in the attribute/structure segmentation for skin lesions. This can be attributed to various challenges such as insufficient annotated data, imbalanced datasets, and the complexity of the structures within each lesion, which exhibit diverse appearances. To address these challenges, many previous approaches have employed standard deep learning models and utilized transfer learning techniques. Transfer learning has proven effective in mitigating data scarcity issues. While there are alternative learning approaches, such as one-shot learning and zero-shot learning, they have not been extensively explored for attribute segmentation in skin lesions.

This section has given a detailed description of the dermoscopic structures present in skin lesions that need to be segmented, and we have discussed how the presence of these attributes could help the dermatologists in making an accurate diagnosis of the skin lesion type. We have also discussed the various challenges present in the skin lesion image dataset, including the similarity among various structures, the presence of more than one structure within one lesion, and the presence of highly imbalanced datasets. We have discussed the need for a CNN-based model in order to perform the structural-level segmentation of skin lesions. A detailed survey of state-of-the-art literature has also been given with regard to the segmentation of skin lesions and the attribute-level segmentation of skin lesions.

The rest of the paper is organized as follows: Section 2 details the existing methodological background of ResNet-50 and UNet architectures. The proposed Dermo-Seg model, along with the hybrid loss function, is discussed in detail in Section 3. The experiments of the proposed model, their results of evaluation, and the details of the dataset are given in Section 4. Section 5 presents the comparisons of our proposed model with existing state-of-the-art techniques and a discussion of their obtained results. Section 6 concludes this paper and discuss future works. References are given at the end of the paper.

## 2. Methodological Background of U-Net and RseNet-50 Architectures

In this section, we discuss the existing U-Net and ResNet-50 architectures.

### 2.1. U-Net Architecture

In this section, we examine the U-Net model that is present in the existing literature. UNet architecture [68] applies the concept of fully convolutional neural networks to effectively capture context-based features and accurate localization. U-Net [68] architecture is made up of two basic components: the encoder and the decoder. These parts create the first and second halves of the model, respectively. The encoder, also known as the contracting path, extracts features from the input image. It applies pre-trained CNN architectures, such as ResNet or VGG, as a backbone. In the U-Net architecture, the encoder employs two consecutive blocks of 3 × 3 convolutions that are unpadded. These are followed by rectified linear unit (ReLU) activation function with 2 × 2 maxpooling and a stride of 2 for down-sampling. This down-sampling process helps to encode the input image into feature map representations at various levels, doubling the number of features at each down-sampling step.

On the other hand, the decoder, also known as the expansive path, constitutes the second part of the U-Net architecture. It aims to reconstruct the segmentation by projecting the discriminative features trained by the encoder into the pixel space, thus achieving higher resolution. Each step in the decoder involves the up-sampling of the feature map, followed by a 2 × 2 up-convolution that reduces the number of feature channels by half. Additionally, the feature maps cropped from the corresponding position in the contracting path are concatenated with the up-sampled feature map. Two consecutive 3 × 3 convolutions, each followed by ReLU activation, are then applied. The U-Net model is made up of a total of 23 convolutional layers. Originally introduced in 2014 for the semantic segmentation of biomedical images, the U-Net model leverages the concept of classification per pixel. It is commonly employed in medical imaging to localize specific parts of an image, typically where abnormalities exist. In semantic segmentation, unlike classification, the input and output maintain the same shape, enabling the precise localization of objects or areas of interest. The architecture of UNet is shown in Figure 3.

The architecture incorporates a 1 × 1 convolutional layer as the last layer, serving the purpose of mapping a feature vector consisting of 64 components to a specific number of classes. In our proposed technique, we replaced the encoder layers of the UNet model with the ResNet architecture, which functions as the backbone of the network.

### 2.2. ResNet Architecture

ResNet [45] was introduced to address the issue of degradation or vanishing gradient that occurs in deeper networks. When deeper networks converge, they tend to degrade, which is not due to overfitting but rather a decline in training accuracy. The ResNet architecture successfully tackles this degradation problem by explicitly mapping the stacked layers to residual mapping. Unlike previous architectures, where the stacked layers directly mapped to the desired underlying mapping, ResNet defines another mapping called $G\left(x\right)=H\left(x\right)-x$, where *H(x)* represents the desired underlying mapping. The stacked non-linear layers are then mapped to this residual mapping. The original mapping can be formulated as $G\left(x\right)+x$. The key hypothesis is that optimizing the residual mapping is easier compared to optimizing an unreferenced mapping. The $G\left(x\right)+x$ formulation is implemented in the feedforward network using “shortcut connections.” The building block of residual learning is given in Figure 4.

The shortcut connections in ResNet enable “identity mappings”, where the outputs of these connections are further added to the outputs of the stacked layers. Importantly, these identity mappings do not introduce additional computational overhead. Deep residual networks, such as ResNet, exhibit improved accuracy as the model depth increases, unlike plain models that experience accuracy degradation with increased depth. In the literature, different versions of ResNet have been introduced, distinguished by the number of layers, such as 18-layer, 34-layer, 50-layer, 101-layer, and 152-layer architectures. In our proposed model, we utilize the ResNet-50 architecture as the backbone in the U-Net model for the task of structure segmentation. The convolutional layers for various parameters are given in Table 2.

## 3. Proposed Methodology

In this section, we have provided a comprehensive description of our novel hybrid model known as Dermo-Seg system, which is based on ResNet-UNet, and designed specifically for the structural segmentation of skin lesions.

### 3.1. Proposed Model Overview

In this section, we have given an overview of our proposed Dermo-Seg model. The attribute-level segmentation of skin cancer lesion is a very challenging task due to the presence of similar or overlapping structures within in them. To meet this challenge, we have proposed a novel attribute segmentation method for skin lesions using the proposed hybrid loss function. The overview of the proposed model is shown in Figure 5. The proposed model is made up of two parts, an encoder and a decoder, which is like the original UNet [68]. The encoder part is replaced by ResNet-50 architecture [45]. The ResNet-50 is used as a backbone in UNet architecture. There are some variations that have been made in the original ResNet-50 architecture to make it work better and more efficiently in our case of skin lesion attribute segmentation... The ResNet-50 encoder part down-samples the input image of the skin lesion. After that, the decoder part up-samples the pixels.

### 3.2. Proposed ResNet-UNet Model

In this proposed work, we have replaced the encoder block of UNet with ResNet architecture to exploit the residual mapping. The proposed architecture of the network is shown in Figure 6. In this, BN denotes batch normalization, MP stands for maxpooling, and BN_ACT refers to batch normalization and ReLU activation. Moreover, the CONV_BLOCK represents the convolution layer, batch normalization, and activation layer, as shown in Figure 7.

In Figure 6, the double arrow represents the zero-padding layer. Figure 6 shows half of the architecture. In total, there are 33 CONV_BLOCKS, but here, due to space constraint, these are limited to CONV_BLOCK 16. The rest of the CONV_BLOCKS follow the same pattern. The two CONV_BLOCKS are connected via zero padding. After every two CONV_BLOCKS, there is an addition part where the previous addition layer is added to the existing layer, as shown in Figure 6. Moreover, at some points, two CONV2D layers are situated after the activation layer. These activation layers are connected to concatenate a portion of the up-sampling layers. After CONV_BLOCK 33, there is a final batch normalization layer and activation layer. After that, the up-sampling section starts. The complete architecture of the up-sampling block is shown in Figure 8. After up-sampling the layer, there is a concatenate layer and two CONV_BLOCKS. All four concatenation layers are concatenating the activation functions from the encoder part to the up-sampling layer of decoder.

### 3.3. Hybrid Loss Function

Although the data imbalance issue was dealt with by down-sampling the maximum class, the positive class ratio was still low. Segmentation is attributed according to pixel classification. This task was one of binary segmentation; the classes belong to two classes: the negative class (black color), and the positive class (white color). The negative class was still dominant over the positive class. This is because the attributes occupy a much smaller area; for example, below is an attribute image of a milia-like cyst. The sample mask of the presence of a milia-like attribute in the skin lesion image is shown in Figure 9.

The positive class is hardly occupying any space; most of the space in this image is occupied by the negative class. An equal number of blank images is also present. So, even after down-sampling the images to positive-class images, the data imbalance issue is still there. This issue is resolved by using the imbalance specific loss function. For our given task, the hybrid loss function is used, which is the combination of the focal Tversky loss and the *IoU* segmentation loss. This hybrid loss function is given in Equation (1):(1)Hybrid loss=focal Tversky loss+IoU segmentation.

Focal Tversky Loss: The focal Tversky loss was introduced [4] to deal with the class imbalance issue. The mathematical representation of the loss is given in Equation (2):(2)$${FTL}_{c}=\sum_{c}{\left(1-{TI}_{c}\right)}^{1/\gamma \;}\;,$$
where the value of γ can be within the range [1, 3], and TI is Tversky Index. TI is the generalization of the dice loss by which the balance between false positive and false negative can be made flexible. The mathematical formula of TI is given in Equation (3):(3)$${TI}_{c}=\frac{\sum_{i=1}^{N}p_{ic}{\;g}_{ic}+\in }{\sum_{i=1}^{N}p_{ic}{g_{ic}+\alpha \;\sum_{i=1}^{N}p}_{i\bar{c}}g_{ic}+\beta \sum_{i=1}^{N}p_{ic}g_{i\bar{c}}+\in \;}\;,$$
where $p_{ic}$ is the probability that pixel *i* belongs to lesion class *c*, and $p_{i\bar{c}}$ is the probability that pixel *i* belongs to non-lesion class $\bar{c}$. The same is true for $g_{ic}$ and $g_{i\bar{c}}$. α and β are hyperparameters that are tuned according to the class imbalance ratio. In the focal Tversky loss, if a certain pixel is misclassified and the value of TI is large, the loss remains unaffected; if the TI value is small and the model misclassified the pixel, the focal Tversky loss decreases. When the value of γ is greater then 1, the less accurate predictions are more focused by FTL. The original work on focal Tversky loss experimented with various values of γ and concluded that 4/3 performed best. In our proposed model, we have used the same value.

*IoU* Segmentation Loss: The *IoU* score is used to evaluate the proposed segmentation model. It measures the similarity of actual pixels to predicted pixels, as given in Equation (4):(4)$$IOU=\frac{TP}{FP+TP+FN}\;,$$
where *TP* is true positive, *FP* is false positive, and *FN* is false negative. However, the *IoU* score is the count-based metric, but the output of the model is usually the probabilities of the pixels belonging to different classes. Therefore, an *IoU* count that approximates the probabilities is proposed by the authors, as given in Equation (5):(5)$$IoU=\frac{I(X)}{U(X)}\;,$$
where I(X) can be approximated using Equation (6):(6)$$I\left(X\right)=\sum_{\upsilon \epsilon V}X_{\upsilon }\times Y_{\upsilon }.$$

Here, X represents the output of the network for the V set of pixels. Y denotes the actual pixels and $Y\;\epsilon \;{\left\{0,\;1\right\}}^{V}$. The formula for approximating U(X) is given in Equation (7):(7)$$U\left(X\right)=\sum_{\upsilon \epsilon V}\left(X_{\upsilon }+Y_{\upsilon }-X_{\upsilon }\times Y_{\upsilon }\right).$$

The IOU loss is defined in the Equation (8):(8)$$L_{IoU}=1-IoU=1-\frac{I\left(X\right)}{U\left(X\right)}\;.$$

The objective function, therefore, is given in Equation (9):(9)$$\arg\underset{w}{min}\;L_{IoU}=1-IoU\;.$$

Equation (9) is solved by employing stochastic gradient descent. The final output after solving the equation is given in Equation (10):(10)$$\left\{\frac{\partial L_{IoU}}{\partial X_{\upsilon }}=\begin{array}{c} \;\;\;\frac{-1}{U\left(X\right)}\;\;\;if{\;Y}_{v}=1\\ \;\;\;\;\;\;\frac{I\left(X\right)}{U{\left(X\right)}^{2}}\;\;\;\;otherwise \end{array}\right\}\;\;,$$
(11)HL=∑c1−TIc1γ+1−IoU=1−I(X)U(X) .

After computation of the gradients, the derivatives can be calculated by incorporating the chain rule. These two loss functions are added together in Equation (11) to make a hybrid loss function. Both loss functions handle the class imbalance problem very effectively. In this section, we have discussed our proposed ResNet-UNet model with hybrid loss function in detail. 

## 4. Experiments and Results

In this section, we document the experiments undertaken to measure the performance of our proposed attribute segmentation model. These experiments are conducted using the ISIC 2018 attribute segmentation Task 2 dataset [52].

### 4.1. Dataset

In our performed experiments, we have employed the ISIC 2018 dataset Task 2 [52], which is specially created for attribute segmentation. It is important to note that this dataset demonstrates a high degree of imbalance. There are 2594 images in total. Each image is associated with different attributes that can be treated as class labels. Thus, each image is marked with five attributes, as given in the ISIC 2018 attribute segmentation dataset. These five attributes are not distributed evenly across the images, as shown in Table 3. The table shows a significant imbalance of attributes present within the dataset of skin lesion images. This data imbalance may potentially introduce bias into the model’s performance. To address this class imbalance problem, we have employed the hybrid loss function as given in Equation (1).

### 4.2. Experimental Setup

We conducted all experiments on Kaggle using the Jupyter framework. First, the dataset was prepared, and the input images were rescaled to a resolution of 512 × 512 to ensure compatibility with the network during training. The entire dataset was then divided into training, validation, and test sets with a ratio of 80:20. For the UNet architecture, we employed ResNet-50 as the backbone model. During our experiments, we set the batch size=8 due to the RAM limitations of Kaggle, as exceeding this value causes crashes. To train the proposed model, we executed it for # of epochs=60 with early stopping criteria. The ADAM optimizer was utilized with a learning rate=0.01. All these parameter settings are given in Table 4. The loss function employed is given by Equation (1). To evaluate the performance of our model, we utilized the Mean Jaccard index or IOU as a measure. The encoder weights used in our model were learned from the Imagenet dataset as they are the best learned weights and have shown excellent performance. The parameters and hyperparameters are fine-tuned to optimize the performance of our proposed model and achieve the best possible results. During the fine-tuning process, specific layers were selectively removed or added to ResNet-50 as an encoder, based on the specific requirements of our model.

### 4.3. Experimental Results and Discussion

To evaluate the model’s performance, the mean intersection over union (IoU) score is used as an evaluation measure. The mean IoU is computed by calculating the intersection over union for each class and then taking the average of these IOU values. The results of the obtained mean IOU scores can be observed in Table 5.

Based on the calculated mean IoU, the performance of the attribute segmentation model for various skin lesion attributes can be summarized as follows.

The proposed ResNet-UNet model achieved a mean IoU of 0.67 for the segmentation of pigment network attribute. It shows that the model has successfully captured the patterns and boundaries of pigment networks within the skin cancer lesions. The negative network attribute achieved a mean IoU of 0.58. Although it is much lower than the pigment network, the proposed architecture still demonstrated reasonable performance in segmenting the presence of negative network patterns within the lesions. The milia-like cysts attribute obtained a mean IoU of 0.53. This shows that the model had moderate success in accurately identifying and segmenting these specific cyst-like structures in skin cancer lesions. The proposed model may encounter challenges in accurately capturing the boundaries and patterns associated with these attributes. The proposed Dermo-Seg model achieved a mean IoU of 0.66 for segmenting the globules within the lesions. This indicates a relatively good performance in accurately identifying and segmenting globular structures, which are important indicators for certain types of skin cancer. The streaks attribute achieved a mean IoU of 0.53. It shows that the model had sufficient success in accurately identifying and segmenting streak-like structures within skin cancer lesions.

The experimental results of the proposed Dermo-Seg model for the structural segmentation of attributes present in skin lesion images, such as pigment networks, negative networks, milia-likes cyst, globules, and streaks, are obtained. The results show the original attribute masks and those predicted by the proposed model. The obtained mask results are shown in Figure 10, Figure 11, Figure 12, Figure 13 and Figure 14. These results show that the proposed framework has performed well, yielding acceptable results in attribute-level segmentation.

Further, the presence of one or more attributes within a lesion indicates the presence of certain types of skin lesion such as melanoma, nevus, BCC, and many more. For example, the presence of pigment network alone or with a milia-like cyst in one lesion suggests the presence of a malignancy [2]. On the other hand, the absence of pigment and the presence of a milia-like cysts in one lesion suggest that we are looking at a benign lesion [2]. So, it is a very sensitive and key factor to observe that the presence or absence of many or one attribute within a single lesion indicates distinct types of skin lesion.

Similarly, the pigment network is a key feature in the diagnosis of melanoma and other pigmented skin lesions. Accurately identifying and segmenting this structure can aid in distinguishing benign lesions from potentially malignant ones.

On the other hand, negative networks are very difficult to detect. These are thin hair-like attributes with a low contrast appearance against the texture of the skin lesion [69]. They are normally light compared to the main color of lesion. Due to this, they may be easily passed over as white colored hairs in the image. This negative network attribute may be present in nevus as well as in melanoma-type skin lesions. It is very difficult to separate nevus from melanoma but the existence of a negative network always indicates melanoma [69]. The negative network attribute is frequently observed in the melanoma-type skin lesion. Our proposed model performed well in detecting this complex negative network attribute even in the presence of the hurdles discussed above. Our model gives IoU = 0.58 for negative network attribute segmentation. In Figure 11, the predicted masks of the negative network are given which suggests the presence of a melanoma-type skin lesion.

Like other skin lesion attributes, it is also very difficult to distinguish between basal cell carcinoma and melanoma if the globules attribute is present in both [69]. These globules may appear at the outer edge or within the skin lesion. Basal cell carcinoma is a type of non-melanocytic skin lesion, and melanoma is a type of melanocytic skin lesion. Both lesions have different origins of occurrence, but the presence of the globules attribute is common in both lesion types. It becomes difficult for practitioners to differentiate between basal cell carcinoma and melanoma as the appearance of the globules attribute is very similar in both types of skin lesion. These globules are dot-like structures. Similarly, streaks attributes can appear at the outer edge of a skin lesion [69], and their presence can potentially indicate different skin lesion types, with a stronger association with melanoma. However, it is important to note that the presence of streaks alone does not provide a definitive diagnosis for melanoma or any other specific skin lesion. To achieve a more accurate diagnosis, it is necessary to consider the presence of other attributes in conjunction with streaks. Streaks are small patterns. From this discussion, it can be observed that most of the skin lesion attributes indicate the presence of melanoma. The most critical is that these attributes also give indication of various other types of skin lesions. This similarity creates hurdles for dermatologists in attempting to differentiate between types based on their attributes. Our proposed model performed well (to some extent) in achieving this goal, as can be observed from the obtained results given in Table 5.

From the above discussion, it can be observed that the performance of the segmentation model in accurately identifying and segmenting dermoscopic structures has direct implications for the diagnostic accuracy of skin cancer types. The mean IOU scores acquired for various structures give an assessment of how well the model has aligned with the ground truth segmentations. Greater mean IoU scores indicate a better association between the predicted and true segmentations, representing improved diagnostic accuracy. Resultantly, this is particularly important in differentiating between benign and malignant lesions, as well as in distinguishing between various other types and sub-types of skin cancer. This is because the inaccurate diagnosis of skin lesion type via consideration of the presence or absence of attributes may lead to a disaster. The training and validation loss curves for five individual learned models for each attribute (five attributes) are shown in Figure 15. The obtained ROC curves for all five attributes are shown in Figure 16. From these ROC curves, it can be observed that our proposed model performs well in the presence of an imbalanced dataset. The computational efficiency of our proposed model is shown in Table 6.

## 5. Discussion

In this section, we discuss and provide a comparison between our proposed Dermo-Seg model and existing attribute segmentation models, including the winner of ISIC 2018 [37,66]. The results of our comparison with IOU (Jaccard Index) are shown in Table 7. Additionally, Table 7 shows a comparison of our proposed structural segmentation model for skin lesions with existing state-of-the-art approaches. We analyzed the performance of various backbone architectures used with U-Net for skin lesion attribute segmentation, specifically with our proposed ResNet50-UNet model, by implying the hybrid loss function. The mean IOU is used as an evaluation measure, and the results are shown in Table 5. Our proposed model shows a robust performance compared to existing state-of-the-art techniques for attribute segmentation of skin lesions.

For example, in terms of segmenting the negative network attribute, our proposed model surpassed the approach presented in [32]. They employed augmentation techniques such as hair occlusion, contrast reduction, and sharpness enhancement to enhance the robustness of their segmentation. Our proposed model achieved a Jaccard similarity or IoU of 0.58 for negative network attribute segmentation, while [32] reported IoU values of 0.149, 0.189, 0.213, and 0.228 were achieved with base networks ResNet151, ResNetv2, DenseNet169, and their proposed ensemble, respectively. Unlike [32], we did not apply any dataset augmentation or pre-processing techniques, yet our proposed method yielded better results by comparison.

Moreover, our method presented significantly better results for attribute segmentation compared to the findings of [29]. In [29], the authors employed b0-EfficientNet as the backbone in conjunction with UNet and LinkNet architectures. Our proposed model also performed well when compared to the attention UNet model presented in [31]. Further, the Dermo-Seg model shows better performance than the attention UNet-based model given in [24]. They trained the transfer learning-based attention UNet on boundary segmentation dataset of skin lesions instead of using the attribute segmentation dataset and used those learned weights as initialization. They achieved IoU scores of 0.535, 0.312, 0.162, 0.187, and 0.197, which are significantly less compared to our Dermo-Seg-achieved scores. Similarly, our proposed model exhibited competitive performance against the second-ranked winner of the ISIC 2018 challenge [66]. The proposed modified PSPNet model achieved IoU scores of 0.482, 0.239, 0.132, 0.225, and 0.145. These scores are significantly worse than our obtained results, scoring more than 0.52 IoU for each attribute. On the other hand, NMN’s method [37], which come forward as the winner of the ISIC 2018 challenge, also shows a very weak performance compared to our proposed Dermo-Seg, although in this case, the authors have used various pre-trained backbone architectures including ResNet variants, DenseNet, EfficientNet, as the encoder and combined the multi-scale information to obtain prediction maps using pyramid pooling methodology. They obtained IoU scores of 0.544, 0.252, 0.165, 0.285, and 0.123 for all five attributes of skin lesions.

From all the existing results for attribute segmentation in skin lesions, it can be observed that the existing state-of-the-art skin lesion attribute segmentation methods have used various backbone architectures. These architectures have many layers, increasing the size of the models. As a result, these architectures have millions of training parameters that require huge computation power and training time. Some of these methods used an ensemble of encoders to extract the feature maps and to concatenate them with the decoder. All these models use many resources, but their obtained results still do not show the desired performance. Some of these methods also applied augmentation techniques to increase the training datasets. Unfortunately, the obtained results showed poor performance in terms of IoU scores. The ISIC 2018 Task 2 dataset is highly imbalanced. These existing models are not performing well on this imbalanced dataset, as can be observed in their results. On the other hand, our proposed Dermo-Seg model shows better and acceptable performances in the attribute segmentation of all five attributes. The proposed model shows IoU scores of more than 0.52 for these attribute-level semantic segmentation. Additionally, the Dermo-Seg model has 9,058,644 training parameters and requires fewer hardware resources, i.e., a 16GB RAM GPU, which is considerably less compared to existing skin lesion attribute segmentation models.

## 6. Conclusions

In this paper, we have discussed and, to a certain extent, resolved the issue of the dermoscopic attribute segmentation of skin lesions. A novel Dermo-Seg system based on ResNet-UNet framework is presented that uses ResNet-50 architecture as a backbone in the UNet model. ADAM optimizer is used to update the weights of the network. To handle the class imbalance problem present in the ISIC 2018 Task 2 attribute segmentation dataset, we have proposed a hybrid of two loss functions, namely, focal Tversky loss and IOU loss functions. By using this proposed loss function, the data imbalance problem is handled to some extent.

The proposed model obtained better results compared to the existing state-of-the-art techniques for skin lesion attribute segmentation. The ResNet 50 architecture extracted useful high-level feature maps that are up-sampled and concatenated in the decoder part. The usefulness of this architecture can be observed in the obtained results of the proposed Dermo-Seg. The proposed model shows a mean IOU score of 0.53 on streaks, 0.67 on pigment networks, 0.66 on globules, 0.58 on negative networks, and 0.53 on milia-like-cysts, which are much better than the existing techniques. The issue of class imbalance, present in the dataset, is resolved by applying a hybrid loss function. This hybrid loss function handled the class imbalance present at the pixel level for each individual attribute. Experiments are conducted to compare the performance of the proposed attribute segmentation model with existing state-of-the-art approaches such as U-Eff (TATL), L-Eff (TATL), attention UNet, LeHealth, NMN, and SANet. It has been shown that the proposed attribute segmentation model provides better segmentation results compared to the competitors in the presence of a class imbalance problem at the pixel level. In the discussion, we also showed that the analysis of dermoscopic structures present in skin cancer lesions provides valuable information for the diagnosis and management of skin cancer. By carefully examining the pigment networks, negative networks, milia-like cysts, globules, and streaks, dermatologists can make informed decisions and differentiate between benign and malignant lesions. The analysis of our obtained results for dermoscopic structures in skin cancer lesions from a medical perspective highlights the importance of the accurate identification and segmentation of these structures for diagnostic accuracy and patient care. This information at the structural level can assist practitioners in accurately determining the class of the skin lesion. The proposed model may serve as a self-explanatory solution for the structure- or attribute-level segmentation of skin lesions. For all we know, our proposed model outperforms existing techniques for skin lesion attribute segmentation.

The obtained results provide insights into the model’s performance for different structures, indicating areas of strength and potential improvement. The achieved IOU scores are around 53% to 67% for all five available attributes. These scores could be further enhanced by proposing state-of-the-art CNN-based strategies. The problem of highly imbalanced datasets needs to be handled by introducing various techniques and by increasing the size of the attribute segmentation datasets.

## Figures and Tables

**Figure 1 diagnostics-13-02924-f001:**
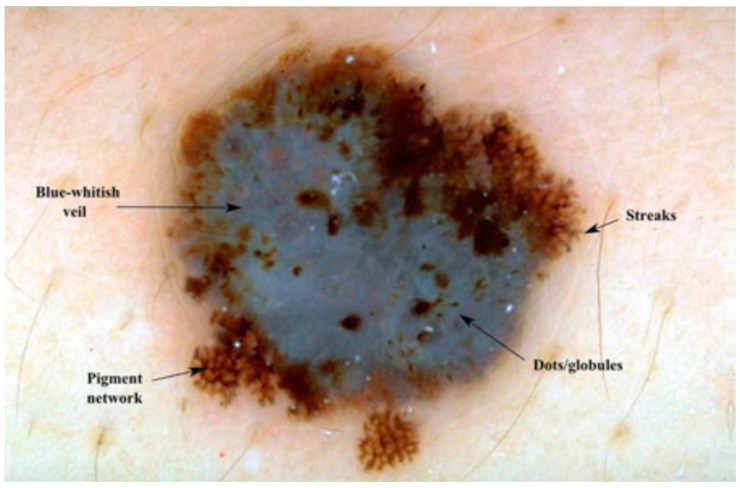
Multiple dermoscopy structures/patterns appear in a skin lesion.

**Figure 2 diagnostics-13-02924-f002:**
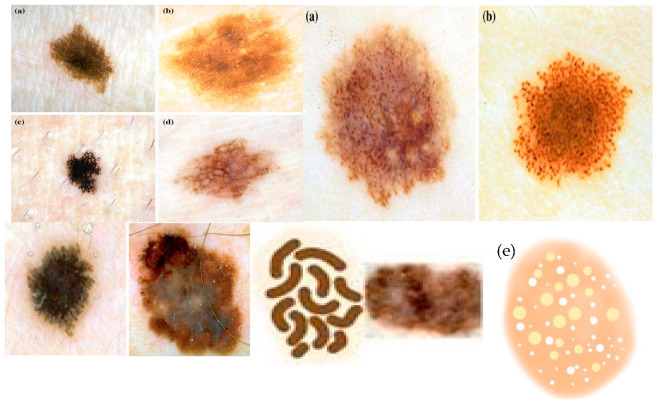
A visual example of five differential structures present in skin lesion: (**a**) pigmented networks; (**b**) globules patterns; (**c**) streaks patterns, (**d**) negative network; (**e**) milia-like cysts.

**Figure 3 diagnostics-13-02924-f003:**
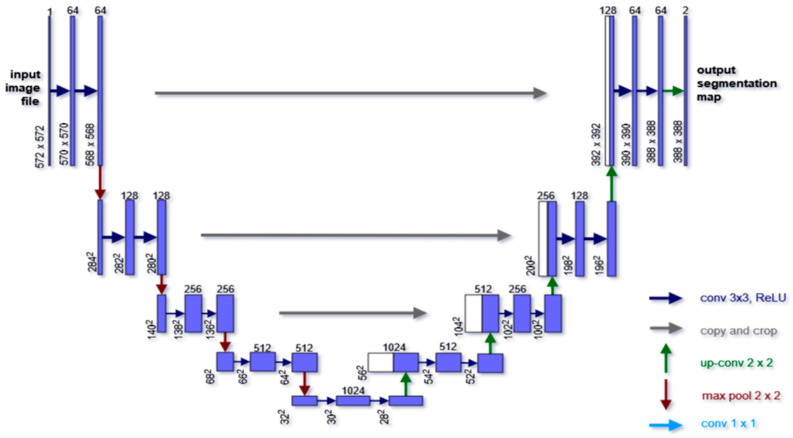
The basic U-Net architecture [68] for image segmentation.

**Figure 4 diagnostics-13-02924-f004:**
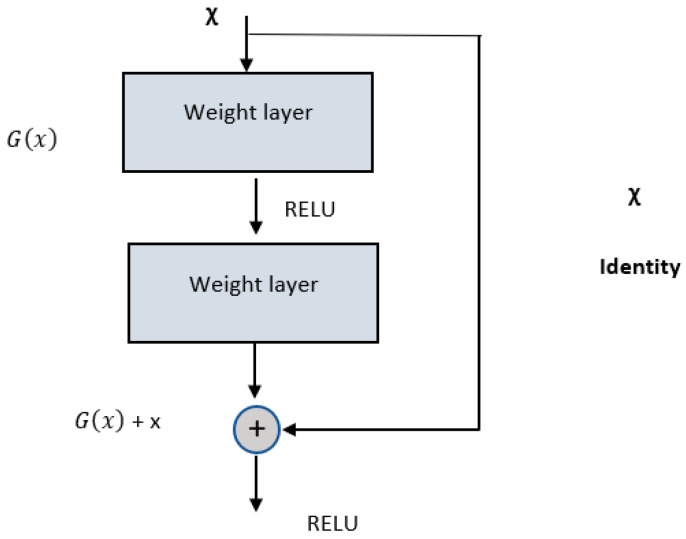
Building block of residual learning.

**Figure 5 diagnostics-13-02924-f005:**
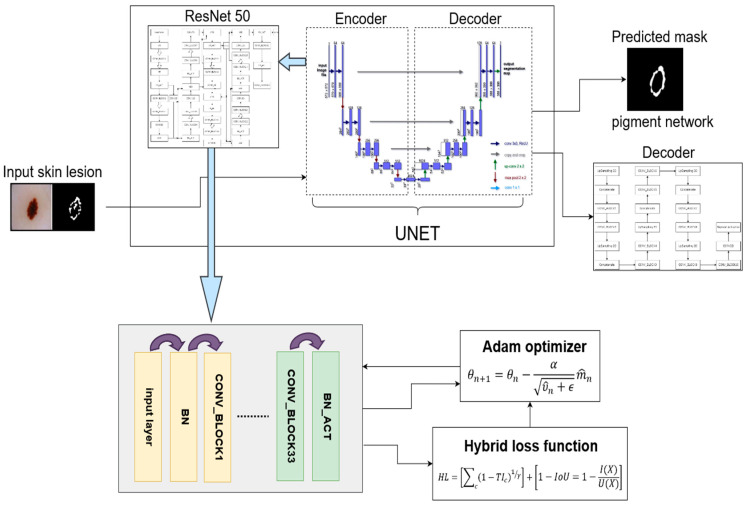
Overview of proposed attributes segmentation model.

**Figure 6 diagnostics-13-02924-f006:**
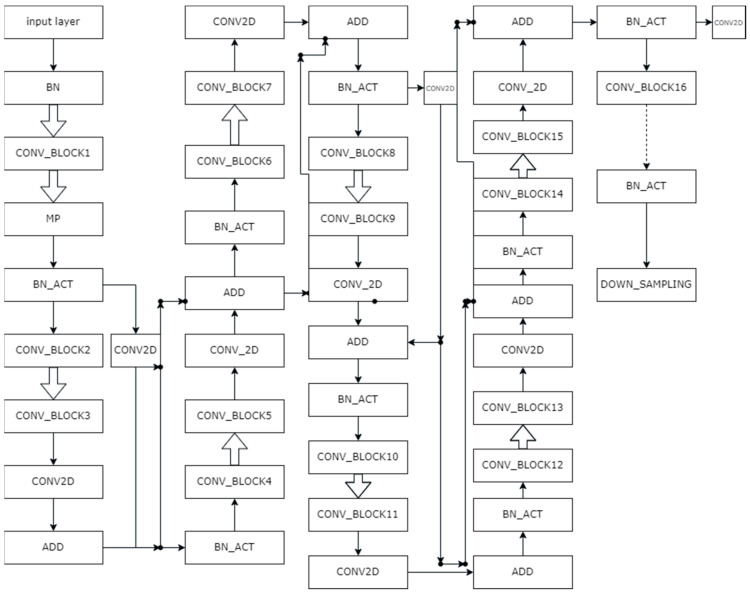
Proposed encoder architecture in ResNet-UNet.

**Figure 7 diagnostics-13-02924-f007:**
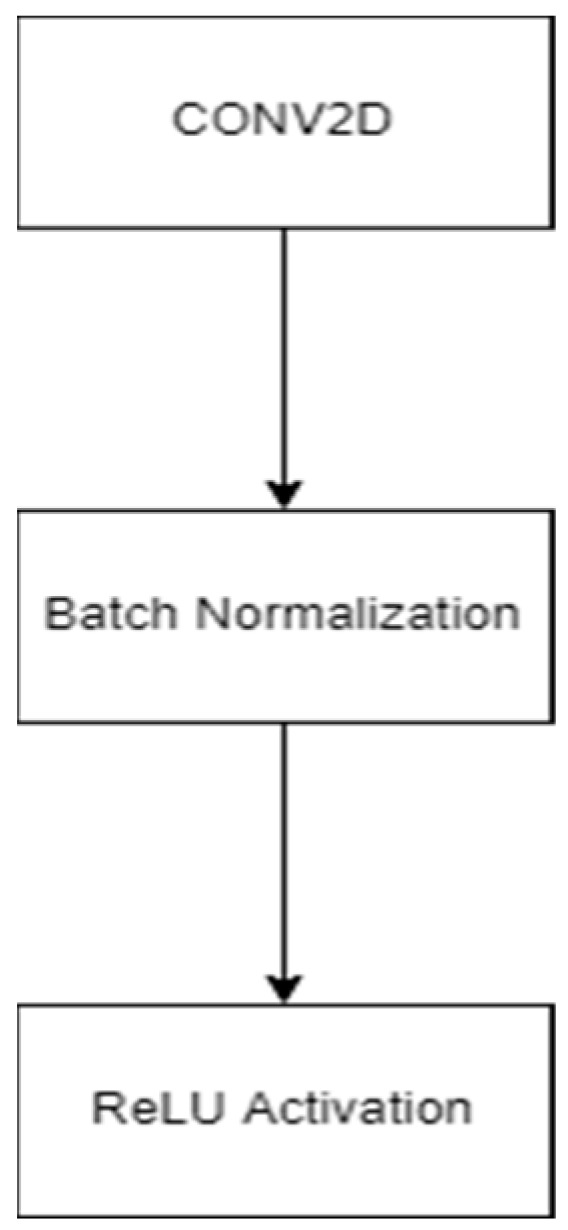
CONV_BLOCK.

**Figure 8 diagnostics-13-02924-f008:**
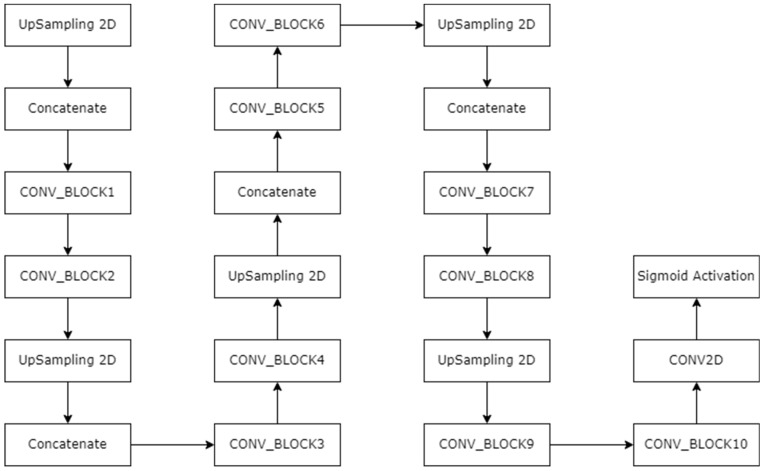
Proposed decoder block in ResNet-UNet architecture.

**Figure 9 diagnostics-13-02924-f009:**
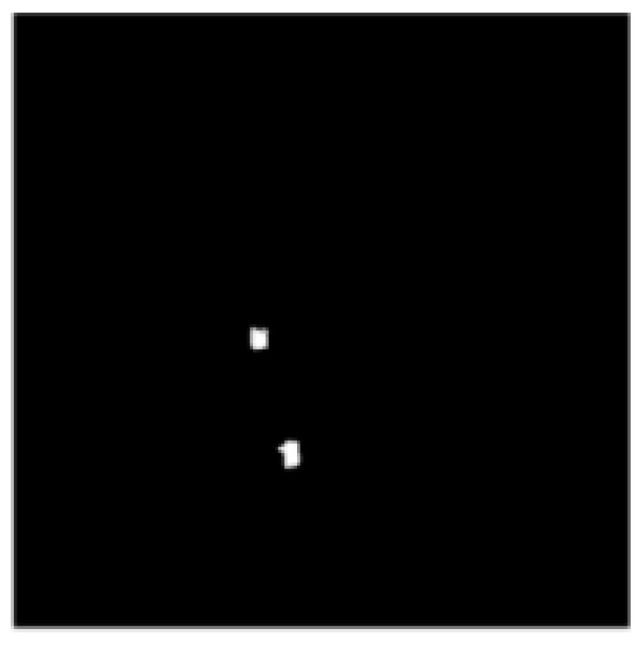
Mask of the presence of a milia-like cyst in the skin lesion image.

**Figure 10 diagnostics-13-02924-f010:**
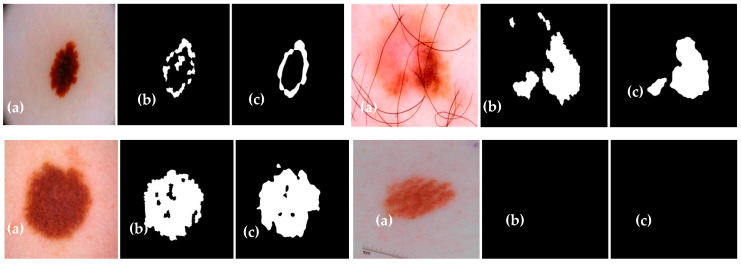
Structural segmentation of the “pigment network” attribute present in a skin lesion image via the proposed ResNet-UNet model: (**a**) original skin lesion image; (**b**) image mask label; (**c**) predicted mask.

**Figure 11 diagnostics-13-02924-f011:**
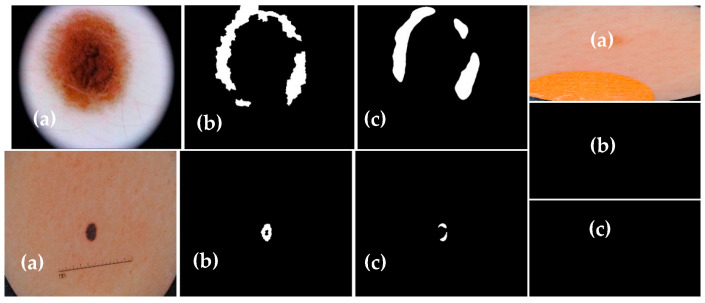
Structural segmentation of the “negative network” attribute present in a skin lesion image via the proposed Res-UNet model: (**a**) original skin lesion image; (**b**) image mask label; (**c**) predicted mask.

**Figure 12 diagnostics-13-02924-f012:**
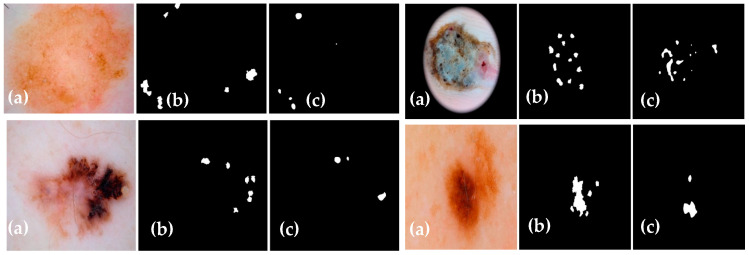
Structural segmentation of the “milia-like cyst” attribute present in a skin lesion image via the proposed Res-UNet model: (**a**) original skin lesion image; (**b**) image mask label; (**c**) predicted mask.

**Figure 13 diagnostics-13-02924-f013:**
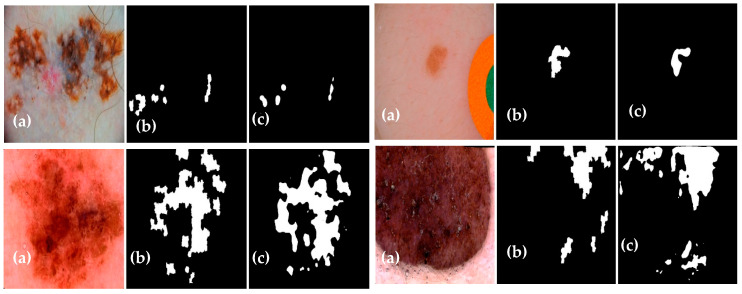
Structural segmentation of the “globules” attribute present in a skin lesion image via the proposed Res-UNet model: (**a**) original skin lesion image; (**b**) image mask label; (**c**) predicted mask.

**Figure 14 diagnostics-13-02924-f014:**
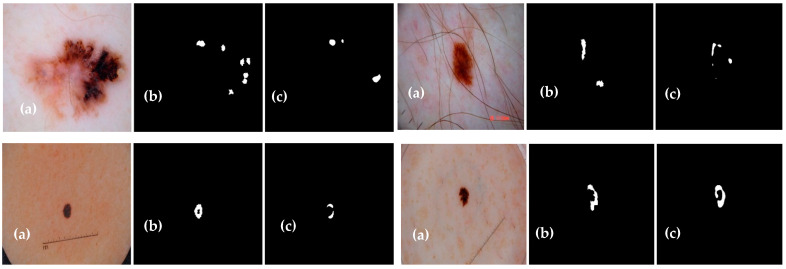
Structural segmentation of the “streaks” attribute present in a skin lesion image via the proposed Res-UNet model: (**a**) original skin lesion image; (**b**) image mask label; (**c**) predicted mask.

**Figure 15 diagnostics-13-02924-f015:**
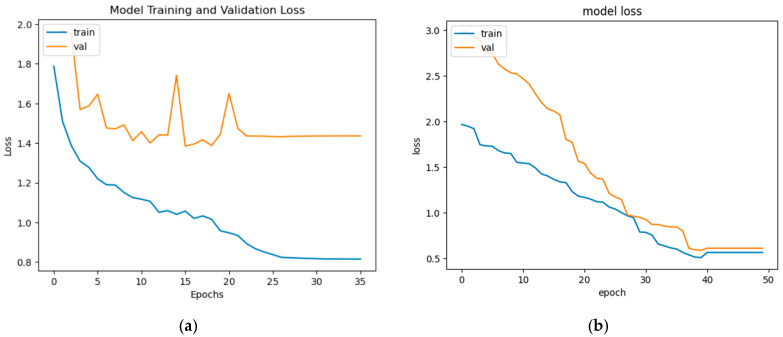
Training and validation loss curves of (**a**) globules; (**b**) pigment network; (**c**) milia-like cysts; (**d**) streaks (**e**) negative attribute.

**Figure 16 diagnostics-13-02924-f016:**
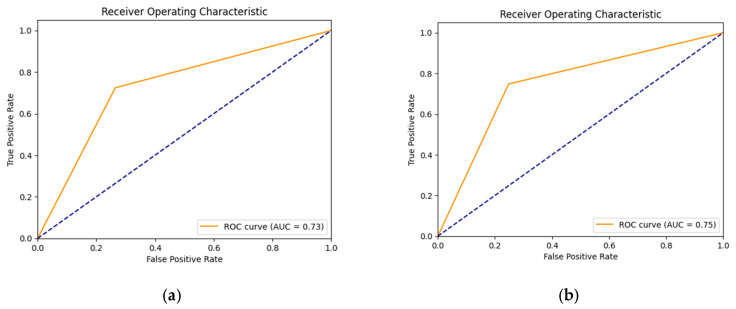
ROC curves of (**a**) globules; (**b**) pigment network; (**c**) milia-like cysts; (**d**) streaks; (**e**) negative attribute.

**Table 1 diagnostics-13-02924-t001:** Comparison of the state-of-the-art methods for segmenting skin lesions attributes.

Cited	Methodology	Dataset	Results	Limitations
[29]	Task-Agnostic Transfer Learning (TATL) using U-Shape with b0-EfficientNet; L-Shape with b0-EfficientNet	ISIC 2018 Task 2	Mean IOU (U-Shape with b0-EfficientNet) Pigment network: 0.565 Globules: 0.373 Milia: 0.157 Negative Network: 0.268 Streaks: 0.243 Mean IOU (L-Shape with b0-EfficientNet) Pigment network: 0.562 Globules: 0.356 Milia: 0.168 Negative Network: 0.292 Streaks: 0.252	Heavy computational resources required; A lot of pre-processing required; Obtained results are very low. i.e., less than 50% of the except pigment network; A very lengthy process followed by the use of various models that consumes resources.
[31]	Attention-UNet model using transfer learning	ISIC 2018 Task 2	Mean IOU Pigment network: 0.535 Globules: 0.312 Milia: 0.162 Negative Network: 0.187 Streaks: 0.197	A lot of pre-processing required.
[32]	Transfer learning-based UNet model with multi-scale convolution (MSC) block pyramid pooling paradigm	ISIC 2018 Task 2	Mean IOU Pigment network: 0.563 Globules: 0.341 Milia: 0.171 Negative Network: 0.228 Streaks: 0.156	A lot of image processing applied including contrast and sharpness adjustment, shrinking and stretching contrast, hair occlusion; Data imbalance handled with augmentation.
[37]	UNet with pyramid pooling paradigm	ISIC 2018 Task 2	Mean IOU Pigment network: 0.544 Globules: 0.252 Milia: 0.165 Negative Network: 0.285 Streaks: 0.123	At test time, various augmentations are applied, and their outputs are merged to predict the final output, which shows dependency and reduces the efficiency of the model.
[66]	Modified pyramid scene-parsing network (modified PSPNet)	ISIC 2018 Task 2	Mean IOU (training) Pigment network: 0.482 Globules: 0.239 Milia: 0.132 Negative Network: 0.225 Streaks: 0.145	They did not show the test data results. Only given training results.

**Table 2 diagnostics-13-02924-t002:** RESNET versions with different layers architecture.

Layer Name	Output Size	18-Layer	34-Layer	50-Layer	101-Layer	152-Layer
conv1	112 × 112	7 × 7, 64, stride 2				
conv2_x	56 × 56	3 × 3 max pool, stride 2				
		$\left[\;\begin{array}{c} 3\times 3,\;64\\ 3\times 3,\;64 \end{array}\;\right]\times 2$	$\left[\begin{array}{c} 3\times 3,\;64\\ 3\times 3,\;64 \end{array}\right]\times 3$	$\left[\begin{array}{c} 1\times 1,\;64\\ 3\times 3,\;64\\ 1\times 1,\;256 \end{array}\right]\times 3$	$\left[\begin{array}{c} 1\times 1,\;64\\ 3\times 3,\;64\\ 1\times 1,\;256 \end{array}\right]\times 3$	$\left[\begin{array}{c} 1\times 1,\;64\\ 3\times 3,\;64\\ 1\times 1,\;256 \end{array}\right]\times 3$
Conv3_x	28 × 28	$\left[\begin{array}{c} 3\times 3,\;128\\ 3\times 3,\;128 \end{array}\right]\times 2$	$\left[\begin{array}{c} 3\times 3,\;128\\ 3\times 3,\;128 \end{array}\right]\times 4$	$\left[\begin{array}{c} 1\times 1,\;128\\ 3\times 3,\;128\\ 1\times 1,\;512 \end{array}\right]\times 4$	$\left[\begin{array}{c} 1\times 1,\;128\\ 3\times 3,\;128\\ 1\times 1,\;512 \end{array}\right]\times 4$	$\left[\begin{array}{c} 1\times 1,\;128\\ 3\times 3,\;128\\ 1\times 1,\;512 \end{array}\right]\times 32$
Conv4_x	14 × 14	$\left[\begin{array}{c} 3\times 3,\;256\\ 3\times 3,\;256 \end{array}\right]\times 2$	$\left[\begin{array}{c} 3\times 3,\;256\\ 3\times 3,\;256 \end{array}\right]\times 6$	$\left[\begin{array}{c} 1\times 1,\;256\\ 3\times 3,\;256\\ 1\times 1,\;1024 \end{array}\right]\times 6$	$\left[\begin{array}{c} 1\times 1,\;256\\ 3\times 3,\;256\\ 1\times 1,\;1024 \end{array}\right]\times 23$	$\left[\begin{array}{c} 1\times 1,\;256\\ 3\times 3,\;256\\ 1\times 1,\;1024 \end{array}\right]\times 36$
Conv5_x	7 × 7	$\left[\begin{array}{c} 3\times 3,\;512\\ 3\times 3,\;512 \end{array}\right]\times 2$	$\left[\begin{array}{c} 3\times 3,\;512\\ 3\times 3,\;512 \end{array}\right]\times 3$	$\left[\begin{array}{c} 1\times 1,\;512\\ 3\times 3,\;512\\ 1\times 1,\;2048 \end{array}\right]\times 3$	$\left[\begin{array}{c} 1\times 1,\;512\\ 3\times 3,\;512\\ 1\times 1,\;2048 \end{array}\right]\times 3$	$\left[\begin{array}{c} 1\times 1,\;512\\ 3\times 3,\;512\\ 1\times 1,\;2048 \end{array}\right]\times 3$
	1 × 1	Average pool, 100-d fc, softmax				
FLOPs		$1.8\times 10^{9}$	$3.6\times 10^{9}$	$3.8\times 10^{9}$	$7.6\times 10^{9}$	$11.3\times 10^{9}$

**Table 3 diagnostics-13-02924-t003:** The distribution of attributes in the ISIC skin lesion segmentation image dataset.

Attributes	No. of Images	% of Images
Streaks	100	2.9%
Pigment Network	1522	58.7%
Globules	602	23.2%
Negative Network	189	7.3%
Milia-like-cysts	681	26.3%
Total images	2594	100%

**Table 4 diagnostics-13-02924-t004:** Training parameters setting of proposed ResNet50-UNet model.

Model Parameters	Values
Image Resolution	512 × 512 × 3
Batch size	8
No. of epochs	60
Learning Rate	0.001
Patience	20
Optimizer	Adam
Loss function	Hybrid
Early Stopping at (Automatic)	
Globules	36
Pigment Network	49
Negative Network	59
Streaks	50
Milia	30

**Table 5 diagnostics-13-02924-t005:** Mean IoU test results of structural segmentation of skin lesion attributes.

Class/Attribute	Mean IoU
Streaks	0.53
Pigment Network	0.67
Globules	0.66
Negative Network	0.58
Milia-like-cysts	0.53

**Table 6 diagnostics-13-02924-t006:** Computational efficiency of proposed Dermo-Seg model.

Factors	Values
Execution Time	02 h 40 min
Total No. of parameters	32,561,114
Trainable parameters	9,058,644
Non-trainable parameters	23,502,470
Space required	16 GB RAM

**Table 7 diagnostics-13-02924-t007:** Comparison of proposed Dermo-Seg with hybrid loss function with state-of-the-art methods, measuring the IOU score.

Method	Pigment Network	Globules	Milia-like Cyst	Negative Network	Streaks
ResNet-151 [25]	0.527	0.304	0.144	0.149	0.125
ResNet-v2 [25]	0.539	0.310	0.159	0.189	0.121
DenseNet-169 [25]	0.538	0.324	0.158	0.213	0.134
b0-EfficientNet [22]	0.554	0.324	0.157	0.213	0.139
U-Eff (TATL) [22]	0.565	0.373	0.157	0.268	0.243
L-Eff (TATL) [22]	0.562	0.356	0.168	0.292	0.252
Ensemble [25]	0.563	0.341	0.171	0.228	0.156
Attention UNet [24]	0.535	0.312	0.162	0.187	0.197
LeHealth method (Second ranked ISIC 2018 challenge) [66]	0.482	0.239	0.132	0.225	0.145
NMN’s method [31]	0.544	0.252	0.165	0.285	0.123
SANet [26]	0.576	0.346	0.251	0.286	0.248
Dermo-Seg	0.67	0.66	0.53	0.58	0.53

## Data Availability

Data can be downloaded from: https://challenge.isic-archive.com/data/#2018 (accessed on 12 October 2022).

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
