# Peer review of "Dermo-Seg: ResNet-UNet Architecture and Hybrid Loss Function for Detection of Differential Patterns to Diagnose Pigmented Skin Lesions"

_diagnostics, 2023, doi:10.3390/diagnostics13182924_

Round 1

Reviewer 1 Report

This paper developed a segmentation network for detecting differential patterns in Dermoscopy Skin images. The segmentation network is constructed by incorporating ResNet layers with UNet. The loss function used in this paper has the ability to capture the class imbalance which comes as a very small fraction of pixels containing patterns. The developed model is experimented with publicly available datasets. I appreciate the effort in doing multiple experiments and comparing the result with the state-of-the-art results as available in the published papers. Unfortunately, I don’t think there is any significant novelty in the proposed methodology. UNet having a residual connection is very common and used in multiple medical image segmentation tasks. 

Can be improved slightly.

Author Response

Original Manuscript ID:  ID: diagnostics-2495449       

Original Article Title: Dermo-Seg: ResNet-UNet Architecture and Hybrid-Loss Function for Detection of Differential Patterns to Diagnosis Pigmented Skin Lesions

To: Editor in Chief,

MDPI, Diagnostics

Re: Response to reviewers

Dear Editor,

Many thanks for insightful comments and suggestions of the referees. Thank you for allowing a resubmission of our manuscript, with an opportunity to address the reviewers’ comments.

We are uploading (a) our point-by-point response to the comments (below) (response to reviewers), (b) an updated manuscript with green, blue, and orange highlighting indicating changes, and (c) a clean updated manuscript without highlights (PDF main document).

By following reviewers’ comments, we made substantial modifications in our paper to improve its clarity, English and readability. In our revised paper, we represent the improved manuscript such as:

(1) Revised Abstract, (2) Revised Introduction, (3) Results section, (4) Discussions and Conclusion sections.

We have made the following modifications as desired by the reviewers:

Best regards,

Corresponding Author,

Dr. Qaisar Abbas (On behalf of authors),

Professor.

Reviewer 2 Report

The manuscript is good however, some minor corrections are required to improve its quality. 

1. The abstract is missing the highest scores achieved in this research. 

2. The introduction should explicitly contain the hypothesis of your research. Provide them in bullet format. 

3. Add the paper's outline at the end of the introduction section. 

4. Section 2 can be added to the introduction section. 

5. Add grids to all diagrams in the manuscript. Figure 15 (a,b,c,d, and e),. Figure 16

6.  Section 6 should be renamed as discussion and discussion should be extended. 

7. Conclusions should be written in the following way: 

1) First paragraph - describing what was done in the manuscript, 

2) Results of the conducted investigation written as answers to hypotheses of your research (hypotheses written in the introduction section)

3) Advantanges and disadvantages of proposed approach. 

4) Directions for the future work 

8. The formulas are part of the sentences which means that if the sentence ends with a formula then put the dot after the formula. If the sentence continues after the formula then put the comma after the formula. 

9. Using the proposed method (Dermo-Seg) you have achieved the highest IOU score (Table 7). However, in your opinion is there any room for improvement? Can this improvement be achieved by artificially enlarging the number of dataset samples (application of generative adversarial networks) ? 

The English language is ok. 

Author Response

(The authors gave the same response as above.)

Reviewer 3 Report

This research addressed an interesting topic related to Diagnosis Pigmented Skin Lesions

The contribution of this work is interesting. However, some comments could be considered.

-        Explore further studies and provide a comparison related to your work, e.g., the strongest points of your work, the contributions and limitations of the previous work concerning your contribution, something like a summarized table.

-        Minimize the abbreviations, and please add a table for them to make the paper easier and increase its readability.

 - Simplify the equations

-        Also, it will be helpful if you provide a summary at the end of each section to conclude your work in the reader's mind.

-        Further discussion of the results

 - Addressing Ensemble Deep Learning Methods in your work, also, You can use and cite  the following reference, which is related to your work:

Towards Comprehensive Chronic Kidney Disease Prediction Based on Ensemble Deep Learning Models

Minor editing of the English language required

Author Response

(The authors gave the same response as above.)

Round 2

Reviewer 1 Report

The paper is improved and can be considered for publication.

Some minor mistakes are there. Careful double-checking is required.